# HUMANNORM: LEARNING NORMAL DIFFUSION MODEL FOR HIGH-QUALITY AND REALISTIC 3D HUMAN GENERATION

## ABSTRACT

Recent text-to-3D methods have marked significant progress in 3D human generation. However, these methods struggle with high-quality generation, resulting in smooth geometry and cartoon-like appearances. In this paper, we found that by fine-tuning the text-to-image diffusion model with normal maps, it can be adapted to a text-to-normal diffusion model, while preserving part of the generation priors learned from large-scale datasets. Therefore, we propose HumanNorm, a novel approach for high-quality and realistic 3D human generation by integrating normal maps into diffusion models. We employ two integration strategies and propose a normal-adapted diffusion model as well as a normal-aligned diffusion model. The normal-adapted diffusion model can generate high-fidelity normal maps corresponding to prompts with view-dependent text. The normal-aligned diffusion model learns to generate color images aligned with the normal maps, thereby transforming physical geometry details into realistic appearance. Leveraging the proposed normal diffusion model, we devise a progressive geometry generation strategy and coarse-to-fine texture generation strategy to enhance the efficiency and robustness of 3D human generation. Comprehensive experiments substantiate our method's ability to generate 3D humans with intricate geometry and realistic appearances, significantly outperforming existing text-to-3D methods in both geometry and texture quality.

## 1 INTRODUCTION

Large-scale generative models have achieved significant breakthroughs in diverse domains, including motion (Tevet et al., 2023), audio (Oord et al., 2018; Agostinelli et al., 2023), and 2D image generation (Rombach et al., 2022; Nichol et al., 2021; Ramesh et al., 2022; 2021; Saharia et al., 2022). However, the pursuit of high-quality 3D content generation following the success of 2D generation poses a novel and meaningful challenge. Within the broader scope of 3D content creation, 3D human generation holds particular significance, given its pivotal role in applications such as AR/VR, holographic communication, and the metaverse. Moreover, it contributes to the advancement of 3D visual perception, understanding, and reconstruction.

To achieve 3D content generation, a straightforward approach is to train generative models like GANs or diffusion models to generate 3D objects using representations such as voxels (Wu et al., 2015) or tri-planes (Chan et al., 2022; An et al., 2023; Wang et al., 2023a). However, these approaches face challenges due to the scarcity of current 3D datasets, resulting in restricted diversity and suboptimal generalization. In the context of 3D human generation, the demands are more strict in terms of diversity and generation quality. Although existing 3D human datasets encompass high-precision scan models, they suffer from limitations in quantity and lack diversity in clothing, poses, head types, and hair variations. This inherent constraint poses a substantial obstacle to the direct generation of high-quality 3D humans.

To overcome these challenges, recent methods (Poole et al., 2023; Lin et al., 2023; Metzer et al., 2023) adopt a 2D-guided approach rather than a direct one to achieve 3D generation. Their core framework builds upon pre-trained text-to-image diffusion models and distills 3D contents from 2D generated images through Score Distillation Sampling (SDS) loss (Poole et al., 2023) and differ-

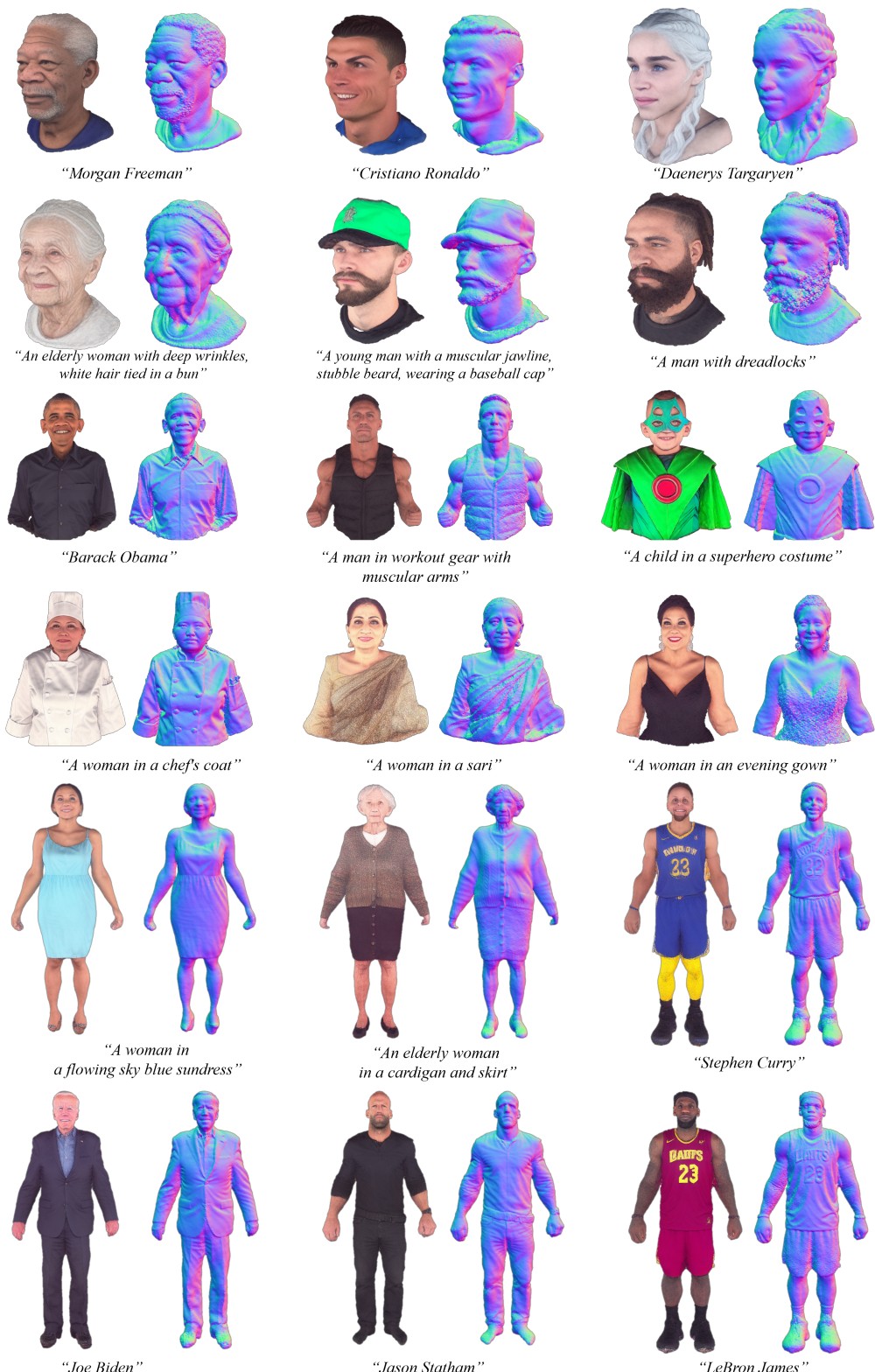

Figure 1: **3D humans generated by HumanNorm from text prompts.** A single view and the corresponding normal map are rendered. **See supplementary for video results**.

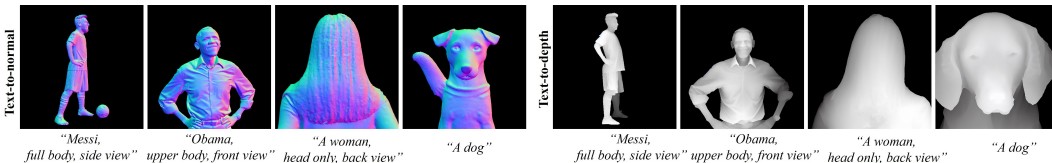

Figure 2: **2D results by normal-adapted and depth-adapted diffusion model.** The view-dependent texts like "front view" are utilized to control the view direction. The texts like "upper body" are employed to manage the body parts generated by the diffusion models.

entiable rendering. Leveraging the image generation priors learned from large-scale datasets, this framework enables more diverse 3D generation. However, current text-to-image diffusion models primarily emphasize the generation of natural RGB images, which results in a limited perception of 3D geometry structure and view direction. This limitation can result in Janus (multi-faced) artifacts and smooth geometry. Although 3D human generation methods (Cao et al., 2023; Kolotouros et al., 2023; Liao et al., 2023) introduce human body models such as SMPL (Loper et al., 2023) and imGHUM (Alldieck et al., 2021) to enhance the quality of human generation, they still fail to address the fundamental limitation of text-to-image diffusion models. This results in sub-optimal geometry with cartoon-like appearances, particularly in areas such as clothing wrinkles.

In this paper, we discovered that the text-to-image diffusion model can be transformed into a text-to-normal diffusion model by fine-tuning it with a small amount of normal maps. Importantly, this process retains a portion of the generation priors learned from large-scale natural images. Building on this, we present HumanNorm, a novel approach for generating high-quality and realistic 3D human models. Specifically, we train a *normal-adapted diffusion model* using multi-view normal maps rendered from 3D human scans and prompts with view-dependent text. Compared with text-to-image diffusion models, the normal-adapted diffusion model filters out the influence of texture and can directly generate high-fidelity surface normal maps corresponding to prompts with view-dependent text. This ensures the generation of 3D geometric details and avoids Janus artifacts. Since normal maps lack depth information, we also learn a depth-adapted diffusion model to further enhance the perception of 3D geometry. The 2D results generated by these diffusion models are presented in Fig. 2. Furthermore, for texture generation, we learn a *normal-aligned diffusion model* from normal-image pairs. This model efficiently integrates human geometric information into the texture generation process. It accounts for elements such as shading caused by geometric folds and aligns the generated texture with surface normal.

Building on the normal diffusion model, we decompose our 3D human generation framework into two components. Initially, we concentrate on generating high-quality geometry by utilizing a progressive optimization approach. Subsequently, guided by the generated geometry, we employ a coarse-to-fine strategy to create realistic textures. The results by HumanNorm are presented in Fig. 1. The key contributions of this paper are: **1)** We introduce normal-adapted diffusion model that can generate normal maps from prompts with view-dependent text, which improves the fundamental ability of 2D diffusion model for 3D human generation. **2)** We learn normal-aligned diffusion model to align the generated texture with surface normal, which transforms physical geometry details into realistic appearances. **3)** We propose a progressive geometry generation strategy for high-quality geometry and a coarse-to-fine texture generation approach for realistic texture.

## 2 RELATED WORK

**Text-to-3D content generation.** Early methods, such as CLIP-Forge (Sanghi et al., 2022), Dream-Fields (Jain et al., 2022), and CLIP-Mesh (Mohammad Khalid et al., 2022), combine a pre-trained CLIP (Radford et al., 2021) model with 3D representations, and generate 3D content under the supervision of CLIP loss. DreamFusion (Poole et al., 2023), a more recent development, builds upon these CLIP-based methods. It introduces the SDS loss and generates NeRF under the supervision of a text-to-image diffusion model. Following this, Magic3D (Lin et al., 2023) proposes a two-stage method that employs both NeRF and mesh representation for high-resolution 3D content generation. Latent-NeRF (Metzer et al., 2023) optimizes NeRF in the latent space using a latent diffusion model to avoided the burden of encoding images. TEXTure (Richardson et al., 2023) introduces a method for texture generation, transfer, and editing. Fantasia3D (Chen et al., 2023a) decomposes the generation process into geometry generation and texture generation to enhance the performance

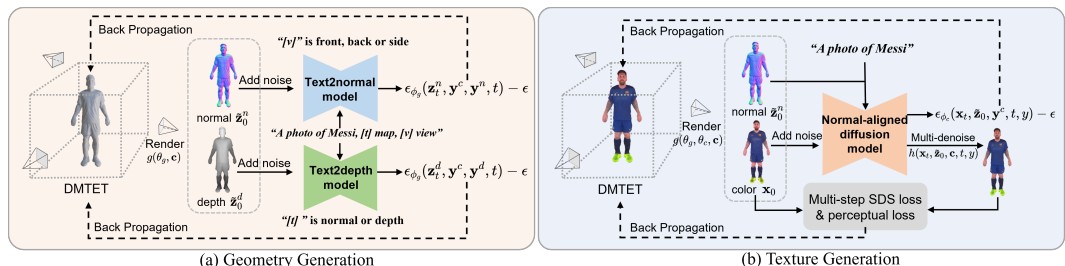

Figure 3: **Overview of HumanNorm.** Our method is designed for high-quality and realistic 3D human generation from given prompts. The whole framework consists of geometry and texture generation. We first propose the normal-adapted and depth-adapted diffusion model for the geometry generation. These two models can guide the rendered normal and depth maps to approach the learned distribution of high-fidelity normal and depth maps through the SDS loss, thereby achieving high-quality geometry generation. In terms of texture generation, we introduce the normal-aligned diffusion model and employ a coarse-to-fine strategy. The normal-aligned diffusion model leverages normal maps as guiding cues to ensure the alignment of the generated texture with geometry. At the coarse level, we exclusively employ the SDS loss, while at the fine level we incorporate the multi-step SDS and perceptual loss to achieve realistic texture generation.

of 3D generation. To address the over-saturated issue, ProlificDreamer (Wang et al., 2023b) proposes a Variational Score Distillation (VSD) loss and produces high-quality and high-fidelity NeRF. IT3D (Chen et al., 2023b) introduces GAN loss and leverages explicitly generated 2D images to enhance the quality of 3D contents. MVDream (Shi et al., 2023) proposes a multi-view diffusion model to generate consistent multi-views for 3D generation. However, all these methods are unable to generate high-quality 3D humans, leading to Janus artifacts and unreasonable body proportions. Although Fantasia3D optimizes geometry and textures separately, it generates geometry using a text-to-image diffusion model, resulting distorted and noisy geometry in some cases.

**Text-to-3D human generation.** AvatarCLIP (Hong et al., 2022) integrates SMPL and Neus (Wang et al., 2021) to create 3D human representations, leveraging CLIP for the supervision of geometry, texture, and animation generation. EVA3D (Hong et al., 2023) introduces a part-based NeRF representation within a GAN-based framework to generate 3D humans. DreamAvatar (Cao et al., 2023) utilizes the pose and shape of the parametric SMPL model as a prior, guiding the generation of humans. In a similar vein, AvatarCraft (Jiang et al., 2023) employs an implicit neural representation with parameterized shape and pose control to generate 3D humans. DreamWaltz (Huang et al., 2023) creates 3D humans using a parametric human body prior, incorporating 3D-consistent occlusion-aware SDS and 3D-aware skeleton conditioning. DreamHuman (Kolotouros et al., 2023) generates animatable 3D humans by introducing a pose-conditioned NeRF model that is learned using imGHUM. AvatarBooth (Zeng et al., 2023) uses dual fine-tuned diffusion models separately for the human face and body, enabling the creation of personalized humans from casually captured face or body images. The most recent model, AvatarVerse (Zhang et al., 2023a), trains a Control-Net with DensePose (Güler et al., 2018) as the control signal to enhance the view consistency of 3D human generation. TADA (Liao et al., 2023) derives a SMPL-X (Pavlakos et al., 2019) body model with a displacement layer and a texture map, using hierarchical rendering with SDS loss to produce 3D humans. While these methods reduce Janus artifacts and unreasonable body shapes by introducing human body models, they still produce 3D humans with cartoon-like appearances and smooth geometry. In contrast, our method is capable of generating intricate geometry and realistic appearances. Additionally, our approach could potentially be applied to other tasks, such as text-to-3D objects since our method does not rely on SMPL or imGHUM models. Finally, several methods, such as DreamFace (Zhang et al., 2023b) and HeadSculpt (Han et al., 2023), primarily focus on 3D head generation. Therefore, they encounter difficulties when generating full-body humans.

## 3 PRELIMINARY

We employ the 2D diffusion-guided approach to achieve high-quality and realistic 3D human generation. In this section, we will introduce the diffusion-guided 3D generation framework and the guidance loss of the diffusion model.

## 3.1 DIFFUSION-GUIDED 3D GENERATION FRAMEWORK

When provided with text $y$ as the generation target, the core of the diffusion-guided 3D generation framework aims to align the images $\mathbf{x}_0$ rendered from the 3D representation $\theta$ with the generated image distribution $p(\mathbf{x}_0|y)$ of the diffusion model. Specifically, during the 3D generation process, the rendered images $\mathbf{x}_0$ are ob-

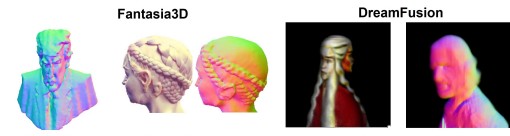

(a) Distorted geometry (b) Unaligned texture (c) Janus artifacts (d) Smooth geometry

Figure 4: **Problems of existing methods.**

tained by randomly sampling cameras $\mathbf{c}$ and rendering through a differentiable rendering function $g(\theta, \mathbf{c})$. Suppose the rendered images from various angles are distributed as $q^\theta(\mathbf{x}_0|y) = \int q^\theta(\mathbf{x}_0|y, \mathbf{c})p(\mathbf{c})d\mathbf{c}$, the optimization objective of diffusion-guided 3D generation framework can be represented as follows:

$$\min_\theta D_{KL}(q^\theta(\mathbf{x}_0|y) \parallel p(\mathbf{x}_0|y)). \tag{1}$$

Directly optimizing this objective is highly challenging, and recent methods have proposed losses such as SDS (Poole et al., 2023) and VSD (Wang et al., 2023b) to solve it. To further enhance the quality of geometry, Fantasia3D (Chen et al., 2023a) proposes to disentangle the geometry $\theta_g$ and appearance $\theta_c$ in the 3D representation $\theta$. In the geometry stage, it utilizes the rendering function $g(\theta_g, \mathbf{c})$ to render normal maps $\mathbf{z}_0^n$ and align the distribution of the rendered normal maps $q^{\theta_g}(\mathbf{z}_0^n|y)$ with $p(\mathbf{x}_0|y)$:

$$\min_{\theta_g} D_{KL}(q^{\theta_g}(\mathbf{z}_0^n|y) \parallel p(\mathbf{x}_0|y)). \tag{2}$$

In the texture stage, Fantasia3D optimizes the texture of 3D objects through Eqn. 1.

**The bottleneck of the diffusion-guided 3D generation.** The bottleneck of the diffusion-guided 3D generation lies in the T2I (text-to-image) diffusion model, which confines itself to parameterizing the probability distribution of natural RGB images, denoted as $p(\mathbf{x}_0|y)$. Therefore, current T2I diffusion model lack the understanding of both view direction and geometry. Consequently, 3D generation directly guided by the T2I diffusion model (Eqn.1) leads to Janus artifacts and low-quality geometry as shown in Fig. 4 (c-d). Although Fantasia3D disentangles geometry and texture, it still encounters issues originating from the T2I diffusion model in both geometry and texture stages. In the geometry stage, directly aligning the rendered normal maps distribution $q^{\theta_g}(\mathbf{z}_0^n|y)$ with the natural images distribution $p(\mathbf{x}_0|y)$ is inappropriate since normal maps significantly differ from RGB images. This alignment results in geometry distortions and artifacts, as depicted in Fig. 4 (a). In the texture stage, minimizing the divergence between $q^{\theta_c}(\mathbf{x}_0|y)$ and $p(\mathbf{x}_0|y)$ leads to the misaligned texture with the geometry as presented in Fig. 4 (b).

## 3.2 GUIDANCE LOSS OF DIFFUSION MODELS

**SDS loss.** SDS loss is widely employed in various diffusion-guided 3D generation frameworks. It translates the optimization objective in Eqn. 1 into the optimization of the divergence between two distributions with diffusion noise, thereby achieving 3D generation:

$$\min_\theta \mathcal{L}_{SDS}(\theta) = \mathbb{E}_{\mathbf{c},t} \left[ (\sigma_t/\alpha_t)\omega(t)D_{KL}(q_t^\theta(\mathbf{x}_t|\mathbf{c}, y) \parallel p_t(\mathbf{x}_t|y)) \right], \tag{3}$$

where $t$ represents the timestep during the diffusion process, and $\mathbf{x}_t$ corresponds to the rendered image $\mathbf{x}_0$ with the noise $\epsilon$ at timestep $t$. $\sigma_t, \alpha_t, \omega(t)$ are the parameters of the diffusion scheduler. Denote $\epsilon_p(\cdot)$ as the pre-trained diffusion model. The gradient of the SDS loss can be computed as follows:

$$\nabla \mathcal{L}_{SDS}(\theta) \approx \mathbb{E}_{\mathbf{c},t,\epsilon} \left[ \omega(t)(\epsilon_p(\mathbf{x}_t, t, y) - \epsilon)\frac{\partial g(\theta, \mathbf{c})}{\partial \theta} \right] \tag{4}$$

**Multi-step SDS and perceptual loss.** Multi-step SDS and perceptual loss are primarily employed for 3D editing and mitigating over-saturation issues in texture generation (Haque et al., 2023; Zhu & Zhuang, 2023). Different from SDS loss, it needs multiple diffusion steps to recover the distribution of RGB images $p_t(\mathbf{x}_0|\mathbf{x}_t, y)$ given $\mathbf{x}_t$ and minimizes the following objective:

$$\min_\theta \mathcal{L}_{multiSDS}(\theta) = \mathbb{E}_{t,\mathbf{c}} \left[ (\sigma_t/\alpha_t)\omega(t)D_{KL}(q_t^\theta(\mathbf{x}_0|\mathbf{c}, y) \parallel p_t(\mathbf{x}_0|\mathbf{x}_t, y)) \right]. \tag{5}$$

Multi-step SDS loss promotes stability during optimization and avoids getting trapped in local optima. To further prevent over-saturation effects, the perceptual loss is also applied to keep the natural style of the rendering images $\mathbf{x}_0$ from $g(\theta, \mathbf{c})$ with the generated images $\hat{\mathbf{x}}_0$ from $p_t(\mathbf{x}_0|\mathbf{x}_t, y)$:

$$\mathcal{L}_p = \mathbb{E}_{\hat{\mathbf{x}}_0 \sim p_t(\mathbf{x}_0|\mathbf{x}_t, y)} \left( \|V(\mathbf{x}_0) - V(\hat{\mathbf{x}}_0)\|_2^2 \right), \tag{6}$$

where $V$ is the first $k$ layers of the VGG network (Simonyan & Zisserman, 2015). Denote the multi-step image generation function of the diffusion model as $h(\mathbf{x}_t, t, y)$, the gradient of the multi-step SDS and perceptual loss can be formalized as follows:

$$\nabla\mathcal{L}_{multistep}(\theta) \approx \mathbb{E}_{\mathbf{c}, t, \epsilon} \left[ \omega(t)(h(\mathbf{x}_t, t, y) - \mathbf{x}_0)\frac{\partial g(\theta, \mathbf{c})}{\partial \theta} \right] + \lambda_p \nabla\mathcal{L}_p. \tag{7}$$

## 4 METHOD

We propose HumanNorm to achieve high-quality and realistic 3D human generation. The whole generation framework of our method has the geometry stage and texture stage, as shown in Fig. 3. We first introduce our normal diffusion model, which consists of the normal-adapted diffusion model and the normal-aligned diffusion model (Sec. 4.1). Then in the geometry stage, based on the normal-adapted diffusion model, we utilize the DMTET as the 3D representation and propose the progressive generation strategy to achieve high-quality geometry generation (Sec. 4.2). In texture stage, building upon the normal-aligned diffusion model, we propose the coarse-to-fine strategy for high-fidelity and realistic appearance generation (Sec. 4.3).

### 4.1 NORMAL DIFFUSION MODEL

In the pursuit of generating a high-quality and realistic 3D human from a given text target $y$, the first challenge lies in achieving precise geometry generation. This entails aligning the distributions of rendered normal maps $q^{\theta_g}(\mathbf{z}_0^n|\mathbf{c}, y)$ from multiple viewports $\mathbf{c}$ with an ideal normal maps distribution $\hat{p}(\mathbf{z}_0^n|\mathbf{c}, y)$. The next challenge is to generate the realistic texture $\theta_c$ while ensuring its coherence with the established geometry $\theta_g$. Therefore, minimizing the divergence between the distribution of rendered images $q^{\theta_c}(\mathbf{x}_0|\mathbf{c}, y)$ and an ideal geometry-aligned images distribution $\hat{p}(\mathbf{x}_0|\mathbf{c}, \theta_g, y)$ becomes essential. The ideal optimization objective is formulated as follows:

$$\min_{\theta_g, \theta_c} \underbrace{D_{KL}(q^{\theta_g}(\mathbf{z}_0^n|\mathbf{c}, y) \parallel \hat{p}(\mathbf{z}_0^n|\mathbf{c}, y))}_{geometry\ generation\ objective} + \underbrace{D_{KL}(q^{\theta_c}(\mathbf{x}_0|\mathbf{c}, y) \parallel \hat{p}(\mathbf{x}_0|\mathbf{c}, \theta_g, y))}_{texture\ generation\ objective}. \tag{8}$$

However, as discussed in Sec. 3.1, the existing T2I (text-to-image) diffusion model is limited to parameterizing the distribution of natural RGB images, denoted as $p(\mathbf{x}_0|y)$, which deviates significantly from the ideal distributions $\hat{p}(\mathbf{z}_0^n|\mathbf{c}, y)$ and $\hat{p}(\mathbf{x}_0|\mathbf{c}, \theta_g, y)$. To bridge this gap, we propose the incorporation of normal maps, representing the 2D perception of human geometry, into the T2I diffusion model to approximate $\hat{p}(\mathbf{z}_0^n|\mathbf{c}, y)$ and $\hat{p}(\mathbf{x}_0|\mathbf{c}, \theta_g, y)$. For the geometry component, we propose to fine-tune the diffusion model, adapting it to generate the distribution of normal map $p(\mathbf{z}_0^n|y)$. In the context of texturing, we utilize ControlNet with normal maps $\mathbf{z}_0^n$ as conditions to guide the diffusion model $p(\mathbf{x}_0|\mathbf{z}_0^n, y)$ in generating normal-aligned images, which ensures that the generated texture aligns with the geometry. The optimization objective incorporating normal maps is defined as follows:

$$\min_{\theta_g, \theta_c} D_{KL}(q^{\theta_g}(\mathbf{z}_0^n|y) \parallel p(\mathbf{z}_0^n|y)) + D_{KL}(q^{\theta_c}(\mathbf{x}_0|y) \parallel p(\mathbf{x}_0|\mathbf{z}_0^n, y)). \tag{9}$$

In addition, we further translate the camera parameters $\mathbf{c}$ into the view-dependent text $\mathbf{y}^c$, serving as an additional condition for the diffusion model. This translation ensures that the generated images align with the view direction, as depicted in Fig. 2. the optimization objective of our method is:

$$\min_{\theta_g, \theta_c} D_{KL}(q^{\theta_g}(\mathbf{z}_0^n|\mathbf{c}, y) \parallel p(\mathbf{z}_0^n|\mathbf{y}^c, y)) + D_{KL}(q^{\theta_c}(\mathbf{x}_0|\mathbf{c}, y) \parallel p(\mathbf{x}_0|\mathbf{z}_0^n, \mathbf{y}^c, y)). \tag{10}$$

Next, we will introduce our 3D human generation framework and construction of the normal-adapted diffusion model and normal-aligned diffusion model used to parameterize $p(\mathbf{z}_0^n|\mathbf{y}^c, y)$ and $p(\mathbf{x}_0|\mathbf{z}_0^n, \mathbf{y}^c, y)$ for geometry and texture generation.

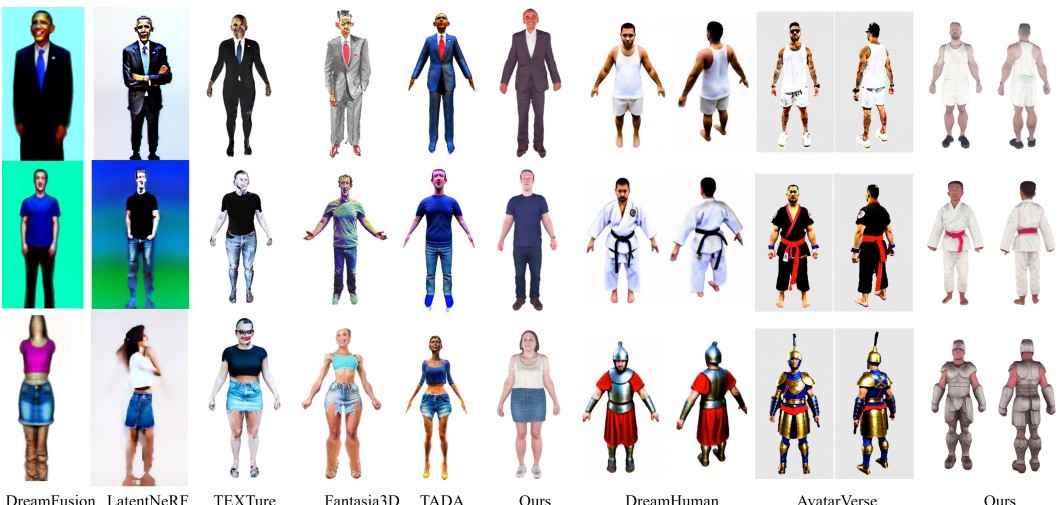

DreamFusion  LatentNeRF  TEXTure  Fantasia3D  TADA  Ours  DreamHuman  AvatarVerse  Ours

Figure 5: **Comparison with text-to-3D content methods and text-to-3D human methods.** The results of DreamFusion are generated by unofficial code. The results of TADA, DreamHuman and AvatarVerse are from their papers and project pages.

## 4.2 NORMAL-ADAPTED DIFFUSION MODEL FOR GEOMETRY GENERATION

Constructing the normal-adapted diffusion model for high-quality geometry generation faces several challenges. First, existing 3D human datasets are scarce, leading to a limited number of normal maps for training. Therefore, we employ a fine-tuning strategy to adapt the text-to-image diffusion models into text-to-normal diffusion model. Then we find the rendered normal maps undergo dramatic changes with variations in viewing angles, which results in potential overfitting or underfitting issues. To mitigate this effects and encourage the diffusion model to focus on perceiving the details of geometry, we transform the normal maps $\mathbf{z}_0^n$ by the rotation $R$ of the camera parameters. The transformed normal maps $\tilde{\mathbf{z}}_0^n$ are used for training of the normal-adapted diffusion model. Furthermore, we also add the view-dependent text $\mathbf{y}^c$ and normal-aware text $\mathbf{y}^n$ (*"normal map"*) as conditions into the diffusion model. The fine-tuning process employs the same optimization objective with the original diffusion model:

$$\min_{\phi_g} \mathbb{E}_{\mathbf{c}\sim p(\mathbf{c}), t\sim \mathcal{U}(0,1), \epsilon\sim\mathcal{N}(\mathbf{0},\mathbf{1})} \left[\|\epsilon_{\phi_g}(\alpha_t\tilde{\mathbf{z}}_0^n + \sigma_t, \mathbf{y}^c, \mathbf{y}^n, t) - \epsilon\|_2^2\right]. \tag{11}$$

The trained normal-adapted diffusion model can guide the geometry generation by normal SDS loss:

$$\nabla\mathcal{L}_{SDS}(\theta_g) = \mathbb{E}_{\mathbf{c},t,\epsilon}\left[\omega(t)\epsilon_{\phi_g}(\tilde{\mathbf{z}}_t^n, \mathbf{y}^c, \mathbf{y}^n, t) - \epsilon)\frac{\partial g(\theta_g, \mathbf{c})}{\partial \theta_g}\right]. \tag{12}$$

In addition to normal SDS loss, we employ several strategies to augment the efficiency and robustness of 3D human generation, outlined as follows:

**Depth SDS loss by depth-adapted diffusion model.** We also finetune a depth-adapted diffusion model by simply changing normal maps to depth maps to calculate depth SDS loss. We found the depth SDS loss can reduce geometry distortion and artifacts in geometry generation.

**DMTET representation and initialization.** We adopt an efficient 3D representation DMTET and initialize it based on a natural body mesh to augment the robustness of 3D human generation. The SDF function in DMTET is further accelerated by the hash encoding of Instant-NGP.

**Progressive Geometry Generation.** We propose a progressive strategy including *progressive hash encoding* and *progressive SDF loss* to mitigate geometric noise and enhance the overall quality of geometry generation. The *progressive hash encoding* employ a mask to suppress high-frequency components of hash encoding for SDF function in DMTET during the initial stage. This allows the network to focus on low-frequency components of geometry and improving the training stability at beginning. As training progresses, we gradually reduce the mask for high-frequency components. Thereby enhancing the details such as clothes wrinkle. For *progressive SDF loss*, we first record the SDF functions $\mathbf{s}(x)$ before reducing the high-frequency mask. Then as training progresses, we

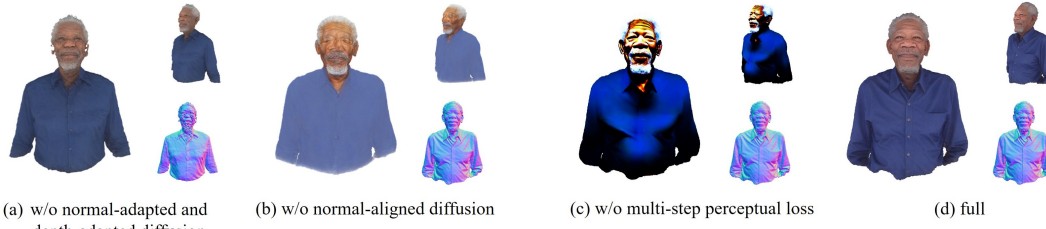

(a) w/o normal-adapted and depth-adapted diffusion     (b) w/o normal-aligned diffusion     (c) w/o multi-step perceptual loss     (d) full

Figure 6: **Ablation studies.** (a-b) Using the text-to-image diffusion model instead of the normal-adapted and depth-adapted diffusion model. (c) Removing the multi-step SDS and perceptual loss.

add the SDF loss to mitigate high-frequency noises or strange geometry deformation: $\mathcal{L}_{SDF}(\theta_g) = \sum_{x \in P} \|\tilde{\mathbf{s}}_{\theta_g}(x) - \mathbf{s}(x)\|_2^2$, where $\tilde{\mathbf{s}}_{\theta_g}(x)$ is the SDF function in DMTET and $P$ is the set of random sampling points.

### 4.3 NORMAL-ALIGNED DIFFUSION MODEL FOR TEXTURE GENERATION

In texture generation, we fix the geometry parameters $\theta_g$ and introduce the normal-aligned diffusion model as guidance. The normal-aligned diffusion model can translate physical geometry details into realistic appearance and ensure the generated texture is aligned with the geometry. Specifically, we employ a ControlNet (Zhang & Agrawala, 2023) to incorporate transformed normal maps $\tilde{\mathbf{z}}_0^n$ as the guided condition of the T2I diffusion model. The training objective of the normal-aligned diffusion model is as follows:

$$\min_{\phi_c} \mathbb{E}_{\mathbf{c} \sim p(\mathbf{c}), t \sim \mathcal{U}(0,1), \epsilon \sim \mathcal{N}(\mathbf{0},\mathbf{1})} \left[ \|\epsilon_{\phi_c}(\alpha_t \mathbf{x}_0 + \sigma_t, \tilde{\mathbf{z}}_0^n, \mathbf{y}^c, t, y) - \epsilon\|_2^2 \right] \tag{13}$$

Then We propose a coarse-to-fine strategy based on the normal-aligned diffusion model.

**Coarse-to-fine Texture Generation.** At the coarse level, we utilize the SDS loss of the normal-aligned diffusion model $\epsilon_{\phi_c}$ for texture generation:

$$\nabla \mathcal{L}_{SDS}(\theta_c) = \mathbb{E}_{\mathbf{c},t,\epsilon} \left[ \omega(t) \epsilon_{\phi_c}(\mathbf{x}_t, \tilde{\mathbf{z}}_0^n, \mathbf{y}^c, t, y) - \epsilon) \frac{\partial g(\theta_c, \mathbf{c})}{\partial \theta_c} \right]. \tag{14}$$

While SDS loss can lead to over-saturated styles and appear less natural as shown in Fig. 6 (c), it efficiently optimizes a reasonable texture as an initial value. We subsequently refine the texture through multi-step SDS and perceptual loss:

$$\nabla \mathcal{L}_{multistep}(\theta_c) \approx \mathbb{E}_{\mathbf{c},t,\epsilon} \left[ \omega(t)(h(\mathbf{x}_t, \tilde{\mathbf{z}}_0^n, \mathbf{y}^c, t, y) - \mathbf{x}_0) \frac{\partial g(\theta_c, \mathbf{c})}{\partial \theta} \right]$$
$$+ \lambda_p \mathbb{E}_{\mathbf{c},t,\epsilon} \left[ (V(h(\mathbf{x}_t, \tilde{\mathbf{z}}_0^n, \mathbf{y}^c, t, y) - V(\mathbf{x}_0)) \frac{\partial V(\mathbf{x}_0)}{\partial \mathbf{x}_0} \frac{\partial g(\theta_c, \mathbf{c})}{\partial \theta_c} \right]. \tag{15}$$

Since the normal-aligned diffusion model executes a complete multi-step diffusion process, the generated images appear more natural and are less prone to oversaturation effects.

## 5 EXPERIMENT

### 5.1 IMPLEMENTATION DETAILS

For each text prompt, our method needs 15K iterations for geometry generation and 10K iterations for texture generation. The entire generation process takes about 2 hours on a single NVIDIA RTX 3090 GPU with 24 GB memory. The final rendered images and videos have a resolution of 1024×1024. More details including dataset, training settings, and others are present in Appendix.

### 5.2 QUALITATIVE EVALUATION

We present qualitative comparisons with text-to-3D content methods (DreamFusion (Poole et al., 2023), LatentNeRF (Metzer et al., 2023), TEXTure (Richardson et al., 2023), and Fantasia3D (Chen et al., 2023a)) and text-to-3D human methods (DreamHuman (Kolotouros et al., 2023), AvatarVerse (Zhang et al., 2023a), and TADA (Liao et al., 2023)).

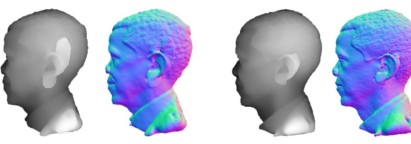

(a) w/o depth SDS          (b) with depth SDS

Figure 7: **Importance of depth SDS.**

**Comparison with text-to-3D content methods.** As illustrated in Fig. 5, the results produced by text-to-3D content methods present a few challenges. The proportions of the generated 3D humans tend to be distorted, and the texture appears to be over-saturated and noisy. DreamFusion struggles to generate full-body humans, often missing the feet, even given a prompt like "the full body of...". In contrast, our method delivers superior results with more accurate geometry and realistic textures.

**Comparison with text-to-3D human methods.** As shown in Fig. 5, text-to-3D human methods yield outcomes with enhanced geometry due to the integration of SMPL and imGHUM human body models. In contrast to these methods, our approach is capable of creating 3D humans with a higher level of geometric detail, such as wrinkles in clothing and distinct facial features. Furthermore, text-to-3D human methods also encounter issues with over-saturation, while our method can generate more realistic colors thanks to our coarse-to-fine texture generation strategy.

### 5.3 ABLATION STUDY

**Effectiveness of normal-adapted and depth-adapted diffusion models.** In Fig. 6 (a), we show the geometry generated by a text-to-image diffusion model instead of our normal-adapted and depth-adapted diffusion models. One can see that the method struggles to generate facial geometry, and holes appear on ears. Additionally, the results display smoother clothing wrinkles and rougher surface. The experiment demonstrates that our normal-adapted and depth-adapted diffusion models are beneficial in generating high-quality geometry.

**Effectiveness of depth SDS.** Existing methods, such as Fanasia3D and TADA, optimize geometry by calculating normal SDS loss. However, we found that only use normal maps as supervision may lead to failed geometry in some regions. As shown in Fig. 7 (a), the ear exhibits artifacts when only using normal SDS loss. This is because the normal of the artifacts is similar to the normal of the head, making it non-salient for the diffusion model. In contrast, we can clearly see the artifacts in the depth map. In Fig. 7 (b), it's evident that the artifacts is reduced when adding depth SDS loss based on our depth-adapted diffusion model, which demonstrates the effectiveness of depth SDS.

**Effectiveness of normal-aligned diffusion model.** In Fig. 6 (b), we experiment with the removal of the normal-aligned diffusion model, opting instead for a text-to-image diffusion model for texture generation. The resulting texture, as can be observed, is somewhat blurry and fails to accurately display geometric details. This is because the text-to-image diffusion model struggle to align the generated texture with geometry. However, using the normal-aligned diffusion model, our method manages to overcome these limitations. It achieves more precise and intricate details, leading to a significant enhancement for the appearance of the 3D humans.

**Effectiveness of coarse-to-fine texture generation.** In Fig. 6 (c), we present coarse stage results with only SDS loss. The generated texture is noticeably over-saturated. However, as shown in Fig. 6 (d). the texture generated through coarse-to-fine strategy exhibits a more realistic and natural color, which underscores the effectiveness of our coarse-to-fine texture generation strategy.

### 6 CONCLUSION

We presented HumanNorm, a novel method for high-quality and realistic 3D human generation by learning the normal diffusion model, which improve the capabilities of 2D diffusion models for 3D human generation. Based on the normal diffusion model, we introduced a diffusion-guided 3D generation framework. Leveraging the proposed normal diffusion model, we devise a progressive geometry and coarse-to-fine texture generation strategy to enhance the efficiency and robustness of 3D human generation. We demonstrated that HumanNorm can generate 3D humans with intricate geometric details and realistic appearances, outperforming existing methods.

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

APPENDIX

## A  3D REPRESENTATIONS

**SDF representation.** Signed Distance Fields (SDF) is a 3D used to describe the geometry surface of an object. It is expressed implicitly through neural networks like MLP. For a sampling point x, everything satisfying $f(x) = 0$ is considered to be part of the object's surface, while the region where $f(x) < 0$ represents the object's interior, and $f(x) > 0$ indicates the object's exterior. SDF can be employed in the synthesis of images from arbitrary viewpoints through methods such as differentiable volume rendering or differentiable Marching Cubes for geometry extraction and re-rendering.

**DMTET representation.** DMTET (Shen et al., 2021) is a hybrid 3D representation that combines explicit and implicit forms. It divides 3D space into dense tetrahedra, which is an explicit partition. Simultaneously, the vertices of these tetrahedra record properties of the 3D object, including SDF, deformation, color, etc. These properties are expressed through the implicit functions of neural networks. By combining explicit and implicit representations, DMTET can be optimized more efficiently and easily transformed into explicit structures like mesh representations. During the generation process, DMTET can be converted into a mesh in a differentiable manner, enabling rapid high-resolution multi-view rendering. We utilize DMTET as the three-dimensional representation in both the geometry generation and texture generation phases.

## B  IMPLEMENTATION DETAILS

**Dataset.** Our dataset comprises 2952 3D human body models. These include 526 models from the THuman2.0 dataset (Yu et al., 2021), 1779 models from the Twindom dataset (TwinDom), and 647 models from the CustomHumans dataset (Ho et al., 2023). We use these models to generate depth maps, normal maps, and color maps. To augment the dataset, we divide the human body into four distinct sections: the head, the upper body, the lower body, and the full body. For each model, we render a set of 120 images, each set comprising depth maps, normal maps, and color maps. The normal maps and depth maps are transformed by the rotation of the camera parameter. We utilize CLIP to generate prompts for the images, supplementing them with additional text to label various data types such as "depth map" and "normal map". We also include descriptors for the view direction, such as "front view", "back view", "left side view", and "right side view", as well as labels for specific regions of the human body, including "head only", "upper body", "lower body", and "full body".

**Training of normal-adapted and depth-adapted diffusion models.** The base stable diffusion model used in our method is SD 1.5. We fine-tune the stable diffusion model using our depth pairs and normal pairs for 15k iterations. The learning rate is set to $1 \times 10^{-5}$ and the batch size is set to 4. Exponential Moving Average (EMA) is used during the training. After fine-tuning, we obtain a normal-adapted diffusion model and a depth-adapted diffusion model. The fine-tuning code is from Diffusers (`https://huggingface.co/docs/diffusers/index`), a library for state-of-the-art pretrained diffusion models for generating images, audio, and even 3D structures of molecules

**Training of normal-aligned diffusion model.** To guide the generation of stable diffusion using a normal map, we load the pre-trained ControlNet based on Stable Diffusion 1.5. We fine-tune the ControlNet for 30K iterations using our image-prompt pairs and take normal maps as control conditions. The learning rate is set to $1 \times 10^{-5}$ and the batch size is set to 4. The fine-tuning code is also from Diffusers.

**Details of progressive hash encoding.** In progressive geometry generation, we employ progressive hash encoding. Specifically, the position encoding for signed distance function (SDF) features has a total of 32 dimensions, where the lower dimensions represent lower-frequency features, higher dimensions represent higher-frequency features. Initially, we utilize a 32-dimensional mask with the first 16 dimensions set to 1 and the latter 16 dimensions set to 0. We multiply this mask with the SDF's position encoding to remove the high-frequency components. During training, every 500 iterations, we convert 2 of the 0 positions in the mask to 1, gradually enabling the network to learn

high-frequency components. After 4,000 iterations, all positions in the mask become 1, resulting in the position encoding encompassing both low-frequency and high-frequency components.

**Details of progressive resolution of marching cube.** We similarly adopt a progressive approach to gradually increase the geometric resolution. Initially, the resolution of the marching cube in 3D space is set to $128^3$. As training proceeds, we incrementally double this resolution every 3,000 iterations. So at 3,000 iterations, the resolution is set to $256^3$, and it will eventually reach $512^3$ at 6,000 iterations. In the early training stages, this results in fewer generated geometry facets, with each facet occupying more pixels in the rendered images. Consequently, the gradients produced by the loss are more evenly distributed across the points of each facet, leading to more stable geometry generation. As the geometric resolution increases, the number of geometry facets also increases, allowing for the representation of more intricate details, including features like hair and clothing folds.

**Details of progressive SDF loss.** During the training process, at the 3,000 iterations, we extract the current geometry to form a coarse mesh. This coarse mesh exhibits the reasonable shape and features a relatively smooth surface. We utilize it to compute the SDF loss for subsequent stages. Specifically, within the bounding box of the 3D generation, we randomly sample 100,000 points at each iteration. Then we calculate the SDF loss by comparing the SDF values of these points in the coarse mesh with the SDF values predicted by the network. The weight of the SDF loss among all the losses is set to 1500 and is only computed after the 3,000 iterations.

**Details of coarse-to-fine strategy.** In the coarse-to-fine strategy for texture generation, the initial 2,000 iterations are utilized as coarse-level optimization and employ SDS loss, while the subsequent 8,000 iterations serve as fine-level optimization, using the multi-step SDS and perceptual loss. For the multi-step loss, the diffusion model performs varying numbers of iterations based on the timestep $t$ with added noise. Specifically, The total timestep of our diffusion model is 1000, when the timestep is $t$, the diffusion model iterates $(t/25+1)$ times. We employ the DPM++ solver as our diffusion scheduler. To enhance training stability, we also incorporate a DU(Dataset Update) strategy similar to what was proposed in instructnerf2nerf. During computation for the multi-step loss at each iteration, we save the image results of multi-step diffusion denoising in a cached dataset, which are reused in subsequent training processes. Every 10 iterations, we will use multi-step diffusion denoising to update the images in the cached dataset.

**Noises and guidance scales of the diffusion model.** In the geometry stage, our text-to-normal diffusion model has a guidance scale of 50, and the text-to-depth diffusion model also has a guidance scale of 50. Similar to the strategy employed in progressive geometry generation, we introduce noise progressively during the geometry stage. In the first 5,000 steps, the timestep $t$ of noise follows the distribution $\mathcal{U}(0.02, 0.8)$. Between 5,000 and 8,000 steps, the timestep $t$ of noise follows the distribution $\mathcal{U}(0.02, k)$ with parameter $k = 0.2 + (0.8 - 0.2)\frac{8000-step}{8000-5000}$. After 8,000 steps, the timestep $t$ of noise follows the distribution $\mathcal{U}(0.02, 0.2)$. In the texture stage, our geometry-guided diffusion model has a guidance scale of 7.5, and the controlled condition scale is set to 1.0. During the coarse level of texture generation, the timestep $t$ of noise follows the distribution $\mathcal{U}(0.02, 0.98)$. In the fine level, the timestep $t$ of noise follows the distribution $\mathcal{U}(0.02, 0.5)$.

**Learning rate and the weight of losses in 3D generation.** We adopt the AdamW optimizer in 3D generation. The learning rate of $\theta_g$ is set to $2 \times 10^{-5}$ and the learning rate of $\theta_c$ is set to $1 \times 10^{-3}$. In the geometry generation, the weight of the normal SDS loss is set to $1.0$ and the weight of the depth SDS loss is $1.0$. In the texture generation, the weight of the color SDS loss is $1.0$ and the weight of the multi-step SDS and perceptual loss are set to $1.0$.

**Part-based optimization.** We primarily divide the human body into four parts for generation: head, upper body, lower body, and the full body. To ensure that the rendered images cover each of these four parts separately, we predefine the camera positions and focal lengths accordingly. During the generation process, the probability of sampling from these four camera positions varies based on the optimization objective. When generating only the head, we sample from the camera capturing the head alone. When generating the upper body of the human, we assign a sampling probability of 0.7 to the upper body and 0.3 to the head. When generating the entire human body, we adjust the sampling strategy progressively. In the first 10,000 iterations, we assign a sampling probability of 0.7 to the entire body and 0.1 to each of the head, upper body, and lower body. In the subsequent

Table 1: **Results of user study.** The table reports the user preference percentages in detail.

| | Q1 (%) | | Q2 (%) | | Q3 (%) | |
|---|---|---|---|---|---|---|
| | Best | Second best | Best | Second best | Most | Second most |
| DreamFusion | 5.36 | 22.27 | 4.73 | 20.55 | 9.27 | 22.55 |
| LatentNeRF | 3.09 | 11.82 | 6.64 | 8.45 | 8.45 | 12.91 |
| TEXTure | 3.64 | 10.27 | 3.91 | 6.64 | 4.91 | 9.09 |
| Fantasia3D | 9.91 | **41.45** | 10.45 | **50.55** | 12.64 | **39.00** |
| Ours | **78.00** | 14.18 | **74.27** | 13.82 | **64.73** | 16.45 |

| | Q1 (%) | Q2 (%) | Q3 (%) |
|---|---|---|---|
| DreamHuman | 23.75% | 16.75% | 38.75% |
| Ours | **76.25%** | **83.25%** | **61.25%** |

5,000 iterations, we assign a sampling probability of 0.1 to the entire body and 0.3 to each of the head, upper body, and lower body.

## C  USER STUDY

Following AvatarVerse (Zhang et al., 2023a), TADA (Liao et al., 2023) and DreamHuman (Kolotouros et al., 2023), we conducted a user study to further assess the quality of the 3D human models generated by our method. Our approach was compared with five state-of-the-art methods across 30 prompts. For each prompt, 50 volunteers (comprising 40 students specializing in computer vision and graphics, and 10 members of the general public) evaluated the color and normal map videos rendered from the generated 3D humans. They voted on three questions:

- Q1: Which 3D human model exhibits the best (and second best) texture quality?

- Q2: Which 3D human model displays the best (and second best) geometric quality?

- Q3: Which 3D human model aligns most closely (and second most closely) with the given prompt?

Since DreamHuman's source code is not publicly accessible, we sourced its results from the project page for a standalone comparison. The results of LatentNeRF, TEXTure, and Fantasia3D are produced using their official code with default settings. Meanwhile, DreamFusion's results were generated using an unofficial implementation in ThreeStudio, a unified framework for 3D content creation (`https://github.com/threestudio-project/threestudio`). We all collect 1500 pairwise comparisons. The results are shown in Tab. 1. One can see that our method surpasses the performance of the four text-to-3D content methods and DreamHuman, particularly in terms of geometry and texture quality. These results underscore the superior performance of our approach.

## D  MORE COMPARISON

We offer further qualitative comparisons between our method and four existing text-to-3D content methods as well as DreamHuman. As depicted in Fig. 11 and Fig. 12, Fantasia3D may generate textures that are not aligned with the geometry (as seen in the second row of Fig. 11). However, the textures produced by our method are accurately aligned with the generated geometry. When compared to the four text-to-3D content methods, our method can generate head-only and upper-body 3D humans with more detailed geometry and a more realistic appearance. In Fig. 13, we present full-body results in comparison with DreamHuman. It is evident that the results produced by DreamHuman contain over-saturated textures and smooth geometry, whereas our method yields a more natural appearance and geometric details.

# E   MORE ABLATION STUDY

**Effectiveness of SDF loss.** In Fig. 8 (a), we display the results obtained in the absence of SDF loss. It is evident that the avatar exhibits a distorted body shape, a consequence of the part-based optimization. However, the introduction of SDF loss effectively constrains the wrong growth of the human body, thereby preventing the formation of such unreasonable body shapes.

**Effectiveness of progressive hash encoding.** In Fig. 8 (b), we conduct an experiment where the frequency of hash encoding is fixed. The results reveal extensive noise on the surface of the geometry, which can be attributed to the high-frequency content learned during the initial training phase. A contrasting case is presented in Figure 8 (c) when a progressive hash encoding approach is employed. Our method significantly reduces the learning of high-frequency information during the initial training phase, resulting in a stable geometry devoid of geometric noise.

# F   EDITING APPLICATIONS

**Text-based Editing.** Our method offers the capability to edit both the texture and geometry of the generated 3D humans by adjusting the input prompt. As demonstrated in Figure 9, we modify the color and style of Messi's clothing, as well as his hairstyle, all while maintaining his identity. While geometry editing poses a greater challenge than texture editing, our method exhibits precise control over geometry generation, even allowing us to generate Messi wearing a hat. Furthermore, the edited geometry is rich in detail, as evidenced by the intricate details in the sweater.

**Pose Editing.** Our method also provides the ability to editing the pose of the generated avatars by adjusting the pose of the initialization mesh and modifying the prompts. The results of pose editing are displayed in Fig 10.

# G   LIMITATIONS

Since we do not utilize human body models such as SMPL and imGHUM, the 3D humans generated by our method are static. Consequently, we are unable to facilitate body and expressive animation using pose and shape parameters. To overcome this limitation, one potential solution could be to integrate the 3D representations with human body models and incorporate pose and shape parameters as additional inputs. This is an avenue we plan to explore in our future work. Furthermore, our method is dependent on the initial geometry. This makes it challenging for us to generate clothing with excessive volume, such as a wedding dress. This is due to the fact that we initialize the geometry with a natural body, devoid of clothing and hair. Lastly, due to the limited size of our training dataset, the training of the geometry-aware diffusion model can be unstable. This may lead to a higher likelihood of overfitting. A potential solution to mitigate this issue is using large model fine-tuning strategies.

# H   ETHICS STATEMENT

The objective of HumanNorm is to equip users with a powerful tool for creating realistic 3D Human models. Our method allows users to generate 3D Humans based on their specific prompts. However, there is a potential risk that these generated models could be misused to deceive viewers. This problem is not unique to our approach but is prevalent in other generative model methodologies. Moreover, it is of paramount importance to give precedence to diversity in terms of gender, race, and culture. As such, it is absolutely essential for current and future research in the field of generative modeling to consistently address and reassess these considerations.

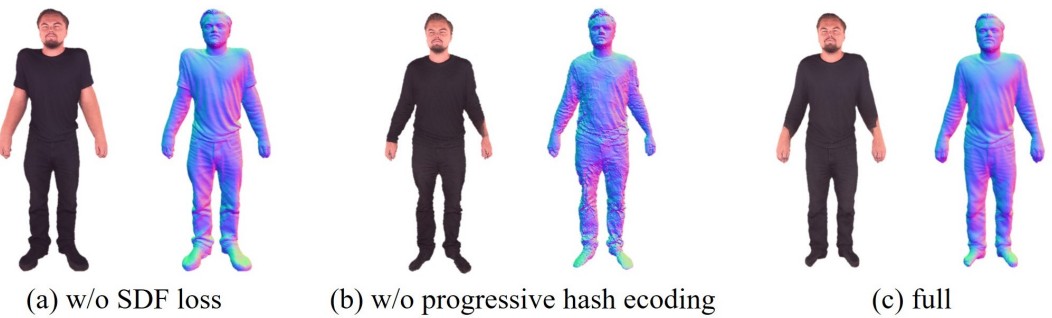

(a) w/o SDF loss          (b) w/o progressive hash ecoding          (c) full

Figure 8: **Importance of SDF loss and progressive hash encoding.**

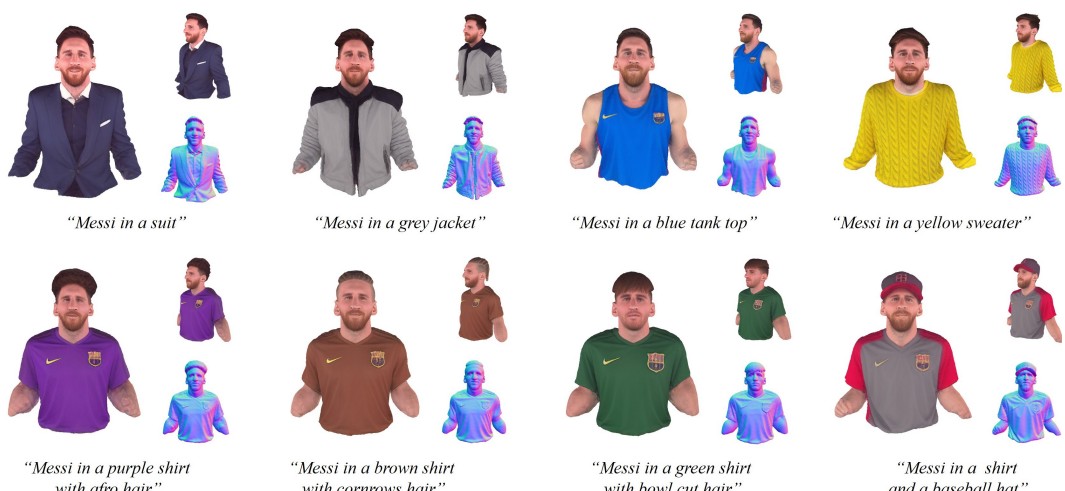

*"Messi in a suit"*          *"Messi in a grey jacket"*          *"Messi in a blue tank top"*          *"Messi in a yellow sweater"*

*"Messi in a purple shirt with afro hair"*     *"Messi in a brown shirt with cornrows hair"*     *"Messi in a green shirt with bowl cut hair"*     *"Messi in a shirt and a baseball hat"*

Figure 9: **Text-based editing.** Our method provides the ability to modify both the texture and geometry of the generated 3D humans by simply altering the input prompt.

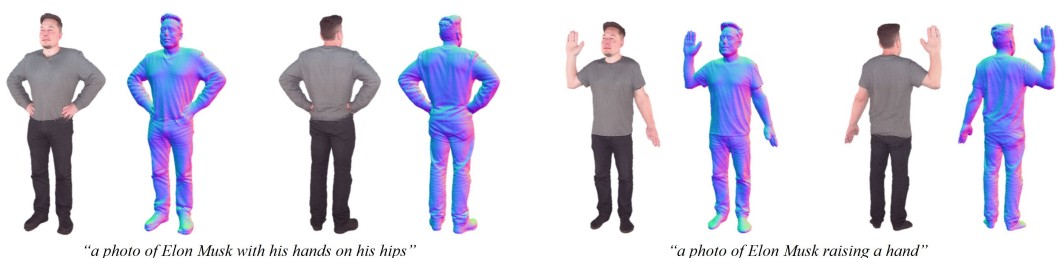

*"a photo of Elon Musk with his hands on his hips"*          *"a photo of Elon Musk raising a hand"*

Figure 10: **Pose editing.** Our method offers the capability to generate 3D humans in various poses by initializing geometry representation with distinct poses.

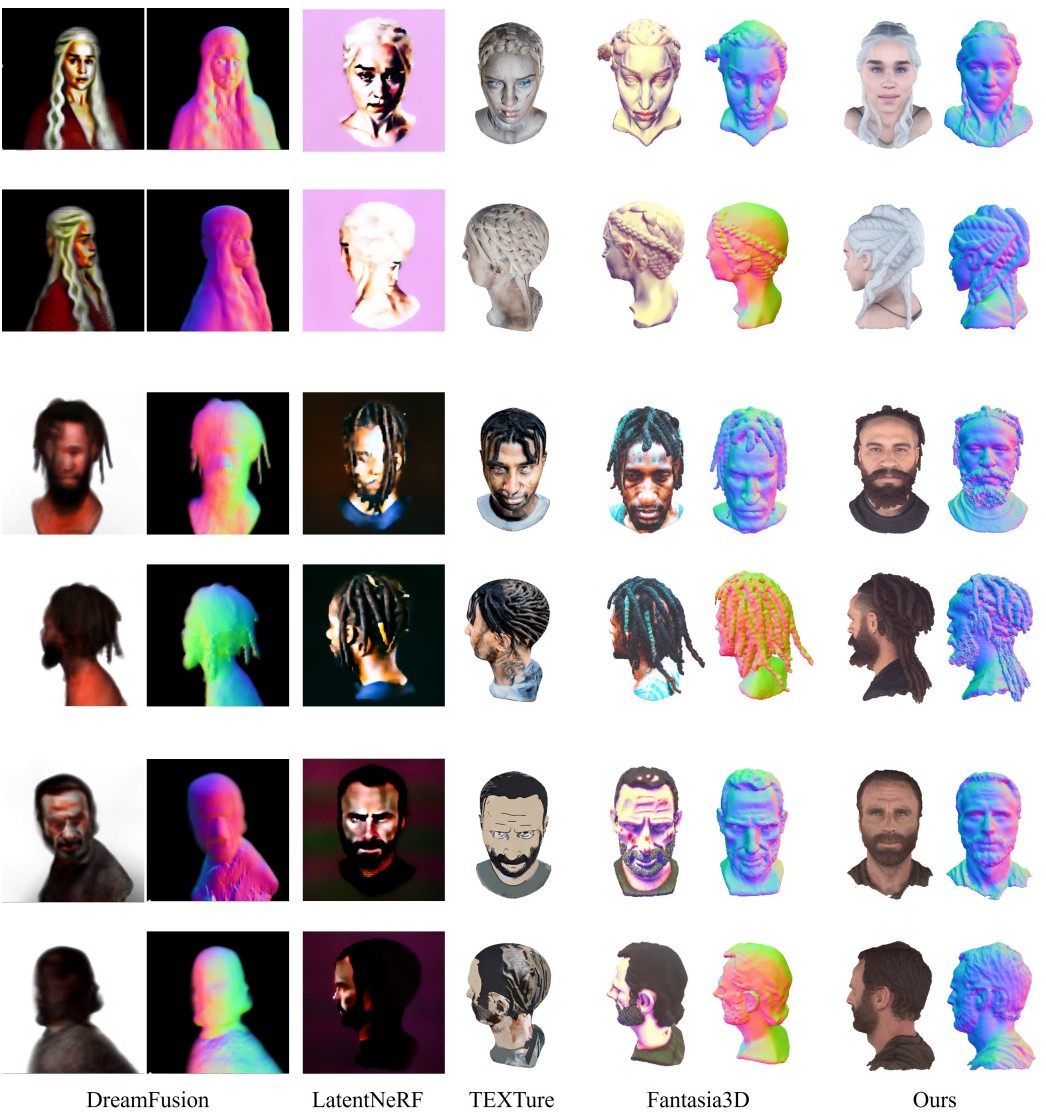

DreamFusion    LatentNeRF    TEXTure    Fantasia3D    Ours

Figure 11: Comparison with text-to-3D content generation methods on the head-only 3D human generation.

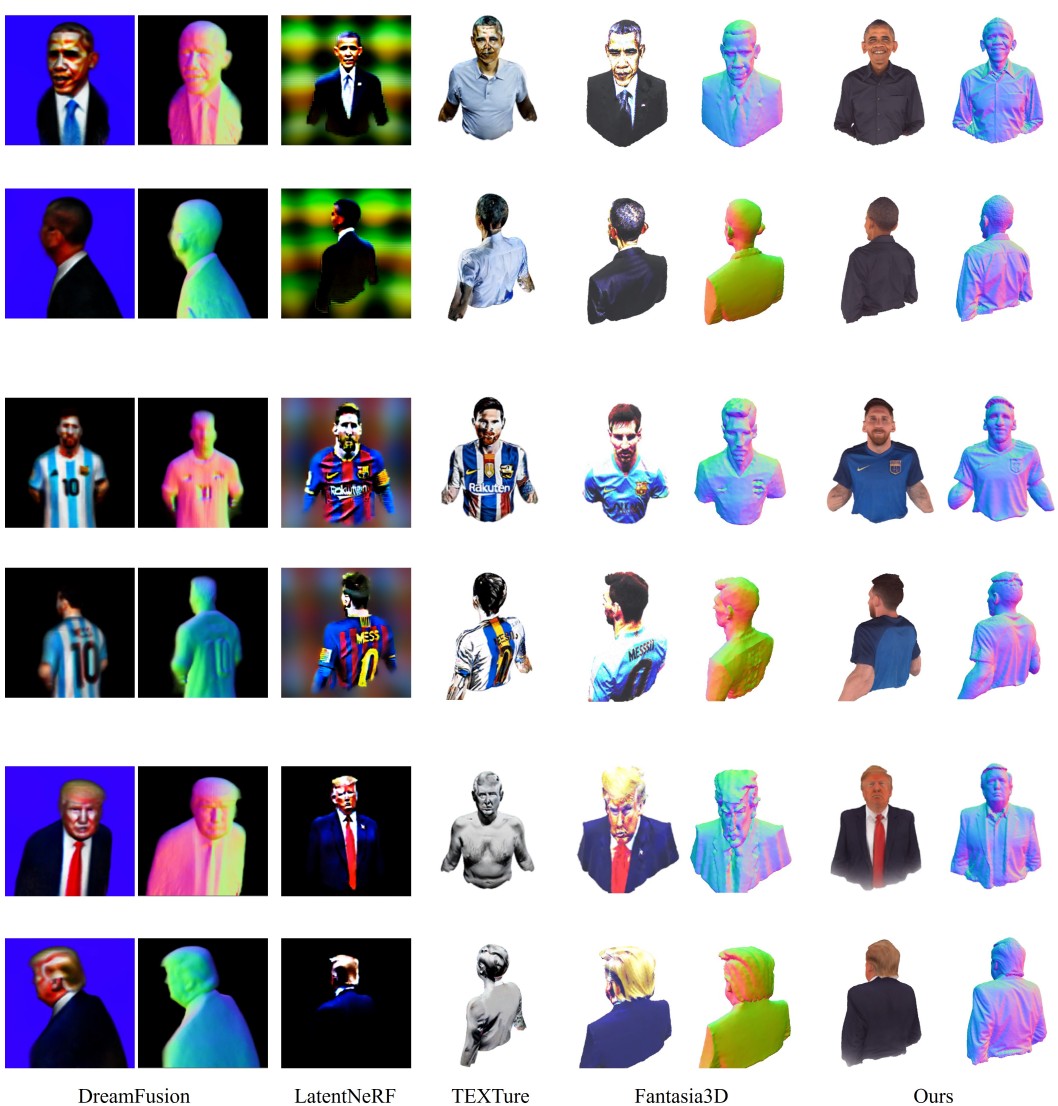

DreamFusion     LatentNeRF     TEXTure     Fantasia3D     Ours

Figure 12: Comparison with text-to-3D content generation methods on the upper-body 3D human generation.

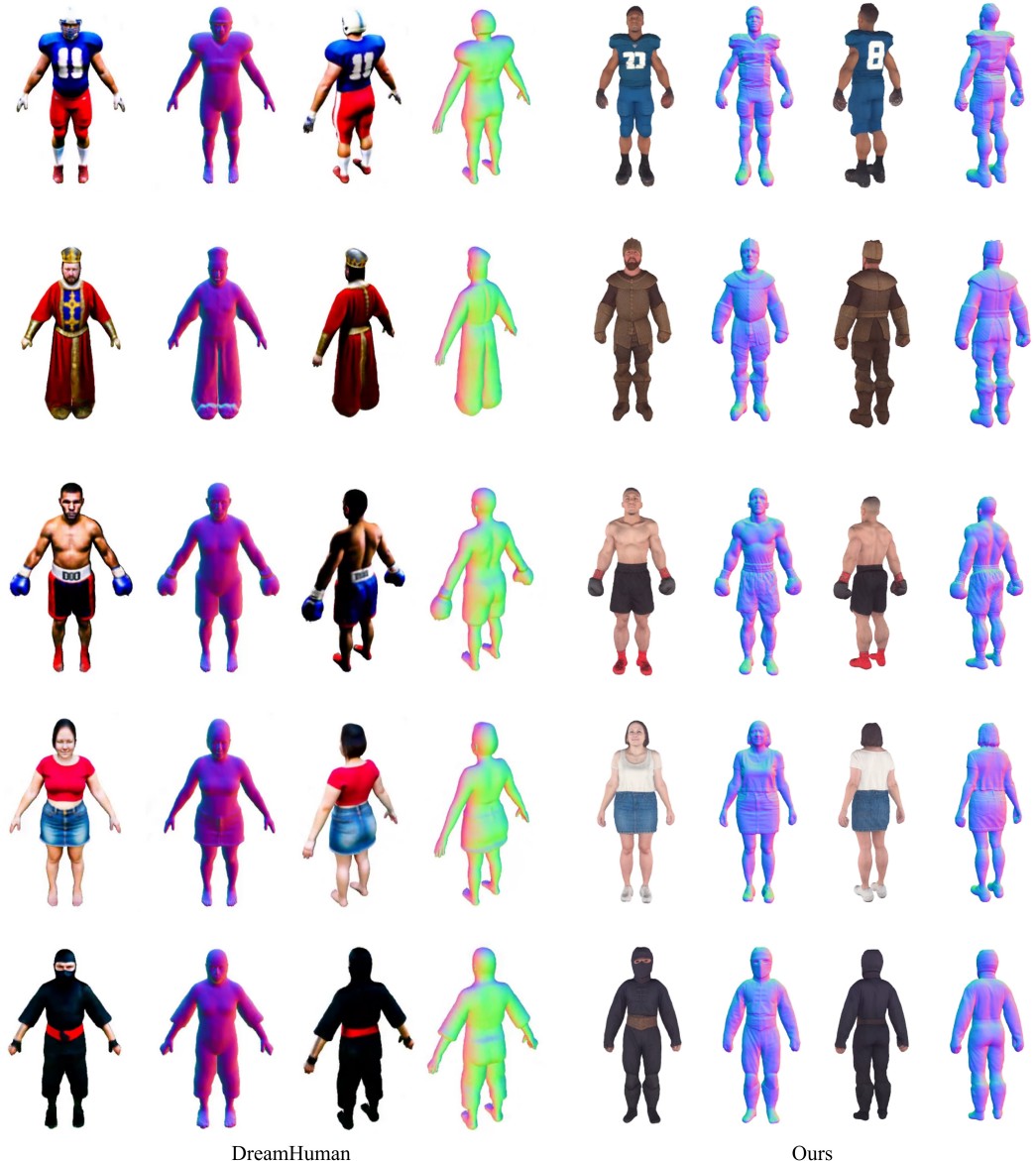

DreamHuman                    Ours

Figure 13: Comparison with DreamHuman on the full-body 3D human generation.

