# OpenReview forum: "HumanNorm: Learning Normal Diffusion Model for High-quality and Realistic 3D Human Generation"
_ICLR.cc/2024/Conference — ICLR 2024 Conference Withdrawn Submission_

### Official Review · Reviewer_LRtj · 2023-10-18

**Soundness:** 2 fair
**Presentation:** 2 fair
**Contribution:** 2 fair
**Rating:** 3
**Confidence:** 5

**Summary:**

In this work, the authors propose a framework for text-to-3d human generation. Specifically, two stages are devised. In the first stage, the authors propose to finetune the text-to-normal and text-to-depth models conditioned on view-dependent prompts, and further use SDS to supervise the DMTet geometry. In the second stage, the normal map is used as controlnet condition to texture SDS. Some tricks like progressive generation and multi-step SDS are used to enhance performance.

**Strengths:**

1. The task of text-to-3d human generation is important to various applications and research.

2. The overall method and pipeline make sense to me in general.

**Weaknesses:**

1. Limited functionality of this pipeline compared to previous studies. Previous text-to-3d human generation works, like AvatarCLIP, DreamAvatar, DreamHuman, TADA, AvatarVerse, all enable the animatable 3d human. That is, they can give a sequence of pose to animated the generated 3d avatars, while this work fails to do this. This hinders further application of this work. Although the authors claim that this method is generalizable to general domains, not limited to human, I don't see any qualitative/quantitative experiments to verify this proof.

Imho, in terms of the general text-to-3d generation, the authors don't present any experiments to verify this. In terms of text-to-3d human generation, this method fails to do the training-free animation.

2. Though finetuning text-to-depth and text-to-normal makes sense to me. Does such isolated training of two models give match results? If not guaranteed, the SDS gradient from depth and normal models could be inconsistent and stochastic, making it hard to normalize.

3. Why not use both depth and normal in the texture generation stage by give to the controlnet?

4. The generated human results still seem blurry to me.

**Questions:**

listed above. The main issue is that, such framework can not animate at inference time, which is inferior to previous baselines and meaningless to industrial use.

---

### Official Review · Reviewer_KiDQ · 2023-10-23

**Soundness:** 2 fair
**Presentation:** 3 good
**Contribution:** 3 good
**Rating:** 5
**Confidence:** 4

**Summary:**

This paper introduces a method to construct 3D geometry and the appearance of human using the data generated from 2D diffusion model. The key idea is to enhance the geometric details of the 3D model by using generated normal and depth. To enable the generation of surface normal and depth, this paper introduces a method to convert the text-to-image model to text-to-normal (or depth). The generated normal and depth maps are further conditioned in the geometry-guided diffusion model that can generate 2D image aligned with geometry. The generated data are used to construct a 3D model with DMTET representation in a progressive and coarse-to-fine framework. Some qualitative results are demonstrated.

**Strengths:**

- The reconstruction results of high-frequency details are quite impressive. I agree that this is possible by the 3D reconstruction supervised with the generated normal maps.
- Showing the feasibility of the idea for converting text-to-image model to text-to-normal is interesting.

**Weaknesses:**

While the overall visual quality seems quite impressive, I have many complaints about experiments and missing details in explanation (which prevents from reproduction of this paper).

[Experiments]
- A research paper should “quantify” the idea. Since it is not a technical report but a research paper, “no quantitative validation” highly suppresses the strength of this paper. While this paper claims a lot of aspects, it only shows the visual results with limited subjects. How can the user study with 50 represent general human perception? This also applies to the ablation study, which requires quantitative evaluation.

-  The validation and demonstration are not aligned with the authors’ claim. The main idea is to conver the 2D image generation model to text-to-normal and normal aligned 2D generation. Validation should reflect the claim. Even for the qualitative results, this paper must show not only 3D human results but also text-to-generation results. Also, many quantitative and qualitative validation needs to be aligned with this claim.


[Description and missing details]
The explanation in the main script is not kind. While the global objective of higher-level idea is quite deliverable, many details are missing. In particular, the details to build a 3D reconstruction model are quite missing; and it was impossible to fully understand “progressive geometry generation” and “coarse-to-fine texture generation”. Please address the questions and concerns in the question section.

**Questions:**

1) In Figure 1, this paper says that it can generate 3D human from text. However, isn’t it reconstruction (from the generated 2D images and surface normal), not directly 3D generation? If so, this sentence is misleading.
2) For training text-to-surface normal with ground truth, how did the author accurately match the training data and associated text prompt of the diffusion model?
3) Other than the front, side, and back views, how could the pipeline generate the images from other views using prompt?
4) In the paper, the authors comment that, the training of generation pipeline takes 2hours. Is that the computational time for “per-subject”? How about other methods? Are their methods also designed for a per-subject reconstruction? Do existing works also require such training time for a specific text?
5) How much time does it take to train per-subject reconstruction?
6) How many views and generated images are required to train the DMTET (for 3D reconstruction) for a specific person?
7) “... we transform the normal maps zn0 by the rotation R of the camera parameters.”: what does it mean?; and how could this mitigate the view-dependent artifacts?
8) How did you apply the multistep perceptual loss in the denoising process?
9) In the section of “Progressive Geometry Generation.”, what is progressive has encoding?; and how do they play an important role in DMTET?; what does it mean by mask? Do you mean by 2D/3D mask? Why is using the mask allowing the network to focus on low-frequency part?
10) How did you compute the camera poses over the generated images?

**Details Of Ethics Concerns:**

-

---

### Official Review · Reviewer_6hVc · 2023-10-26

**Soundness:** 2 fair
**Presentation:** 2 fair
**Contribution:** 2 fair
**Rating:** 5
**Confidence:** 3

**Summary:**

This paper proposes HumanNorm, incorporating normal and depth maps to the text-to-human generation. Experiments shows that HumanNorm outperforms existing text-to-3D methods in both geometry and texture quality.

**Strengths:**

1. I quite appreciate that authors explore the use of normal and depth maps into 2D diffusion models for text-to-human generation. This indeed involves the quality of generated 3D human.
2. Authors provide lots of figures in the paper to show visual quality results. The detailed experimental settings are also provided.

**Weaknesses:**

1. For normal and depth maps generation, authors propose to finetune the 2D diffusion model. As the normal and depth information is the key to learn the details of 3D human, I am wondering if you directly use images from 2D diffusion model to predict the corresponding SMPL/SMPLX and use this as the supervison to learn the normal and depth, will you get similar results as shown in the paper? One advantage of SMPL/SMPLX is that it can represent the characteristics of humans, e.g., human's hand will have 5 fingers. It seems that you methods learns misleading results for hand (with 6 fingers).
2. In your paper, you mention "transform the normal maps z_0^n by the rotation R of the camera parameters". Could you explain how do you rotate the normal maps?
3. When compared with DreamHuman and Avtarverse, it seems that the visual performance improvement is not large. Could authors give more detailed comparison?
4. Authors miss some references on 3D Human generation, e.g.,
1). Efficient 3D Articulated Human Generation with Layered Surface Volumes.
2). Layer-wise 3D Human Generation with Diffusion Model.
I strongly suggest authors checking the paper list on https://github.com/weihaox/awesome-digital-human.
5. Although it is good to involve normal and depths into text-to-human generation, the overall technical contribution is not very high.  If authors could propose some insights on designing model architectures or training strategy based on the observations from experiments, it would add more strength to this paper.

I will consider improving my rating if my above concerns are resolved.

**Questions:**

Questions are listed above.

---

### Official Review · Reviewer_gv3j · 2023-10-31

**Soundness:** 3 good
**Presentation:** 3 good
**Contribution:** 3 good
**Rating:** 6
**Confidence:** 3

**Summary:**

This paper presents a pipeline named HumanNorm for high-quality and realistic 3D human generation. The HumanNorm approach employs a normal-adapted diffusion model and a normal-aligned diffusion model to generate high-fidelity normal maps corresponding to prompts with view-dependent text and to learn to generate color images aligned with the normal maps. The paper also proposes a progressive geometry generation strategy for high-quality geometry and a coarse-to-fine texture generation approach for realistic texture. The results demonstrate that HumanNorm significantly improves both geometry and texture quality, and outperforms previous text-to-3D methods.

**Strengths:**

This paper modifies SDS loss for normal and depth map generation. The normal map used here could eliminate the Janus Artifacts, a vital problem in 3D human generation. It fine-tunes the diffusion model with extra realistic human data on normal, depth, and color map generation. This avoids the limitation of previous methods' cartoon-style results, which suffer from the pre-trained diffusion model distribution.
Generally, this paper is good, though some of the contributions are similar to previous papers.

**Weaknesses:**

This modification of SDS loss seems trivial, and the pipeline is kind of traditional.
The contributions like coarse to fine generation, and view-dependent text prompt is also usual in previous papers.
I have not found any solutions for de-reflection since the THuman2 and other 3D human scan datasets have some problems in surface lighting. The reflection of human skin and clothes are different. Maybe a method to decouple environmental lights from the albedo color should be used.

**Questions:**

Do you have any thoughts on decoupling environmental lights from the albedo color? I think the reflection of human skin and clothes are different.